



# Characterization of Titan Dome, East Antarctica, and potential as an ice core target

Lucas H. Beem[1], Duncan A. Young[2], Jamin S. Greenbaum[2], Donald D. Blankenship[2], Jingxue Guo[3], and Sun Bo[3]

[1]Department of Earth Sciences, Montana State University, Bozeman, MT 59717, USA.
[2]Institute for Geophysics, University of Texas at Austin, Austin, TX 78758, USA
[3]Polar Research Institute of China, Shanghai 200136, China

**Correspondence:** Lucas Beem (lucas.beem@montana.edu)

**Abstract.** Titan Dome is located about 200 km from the South Pole along the 180° meridian within the East Antarctic Ice Sheet. Based on sparse data, it is a region that is identified as having a higher probability of containing ice that would capture the middle Pleistocene transition (1.25 to 0.7 Ma) as a paleoclimate proxy. New aerial geophysical observations collected over Titan Dome were used to characterize the region and assess its suitability as a paleoclimate ice core site. The radar coupled with

an available ice core age model enabled the tracing of isochronal layers throughout the region which also served as constraints on basal ice age modeling. The results of the survey revealed new basal topographic detail, constrained the location of Titan Dome, which differs between community datasets, and suggests that the basal ice beneath Titan Dome is too young to be relevant to study of the middle Pleistocene transition.

## 1 Introduction

The ice domes and ridges of Antarctica hold records of past ice sheet and climate evolution and there is an ongoing international effort (Fischer et al., 2013) to find suitable ice core drilling sites that will have an interpretable climate record that spans the middle Pleistocene transition, dated to between 1.25 Ma and 0.7 Ma (Clark et al., 2006). During this period, marine oxygen isotope records indicate a transition in major ice volume and climate cycles from a predominately ∼41,000 obliquity driven

periodicity to a ∼100,000 periodicity. An ice core's proxies, e.g. trapped atmospheric gases and ice isotopic chemistry, hold records of atmospheric and ice sheet configuration that are key to understanding this transition and climate dynamics more generally.

     Identifying suitable coring locations has primarily been the result of ice dynamic and ice temperature modeling efforts that find regions where ice is dynamically and thermodynamically stable enough to allow for both ice survival for 1.5 Ma and the

existence of a simple chronological record. One such effort used a one-dimensional thermodynamic model to find where the bed is sufficiently cold to prevent present-day basal melting (Van Liefferinge and Pattyn, 2013). With additional criteria of




present-day slow flow of less than 2 m yr$^{-1}$ and ice thickness greater than 2000 m (Fischer et al., 2013) regions with increased likelihood for the recovery of a suitably old ice core were defined (fig. 1).

Not all relevant processes have been explicitly considered in site determination efforts. Additional processes that might im-
pact the existence or quality of the desired ice core include past ice flow reorganization and/or ice divide migration (Beem et al., 2017), subglacial groundwater flow (Gooch et al., 2016), accumulation rate variability, ice surface wind erosion, heterogenous geothermal flux (Jordan et al., 2018). Modeling to enable core site determination has also been hindered by poorly constrained and increasingly divergent estimates of continental scale geothermal flux variability beneath the Antarctic Ice Sheet (Shapiro and Ritzwoller, 2004; Maule et al., 2005; Purucker, 2013; An et al., 2015; Martos et al., 2017). Without geophysical observa-
tions and in some cases direct access, the presence or significance of these processes cannot be determined. Aerial and ground geophysical surveys have occurred for some high probability coring targets, including at Dome C of East Antarctica (Young et al., 2017). Dome C is a leading contender for successful extraction of a sufficiently old ice core due to the characterization of the region (Young et al., 2017), the existence of a proximal ~800,000 year old EPICA ice core (Augustin et al., 2004), and promising ice age modeling (Parrenin et al., 2017). However, finding additional targets remains of interest to enable the
possibility of multiple correlatable cores and the examination of spatial heterogeneity in climate processes.

Titan Dome, located about 200 km along the ~180° meridian from South Pole, is a region that was previously identified as a contender for possible 1.5 million year old ice (Van Liefferinge and Pattyn, 2013). In 2016 and 2017, a partnership between the University of Texas Institute for Geophysics and the Polar Research Institute of China surveyed the South Pole Corridor (SPC) grid over the region to evaluate the location as an ice core target. The existence of an ice core age model at South Pole
(Casey et al., 2014), plus previously collected aerial-geophysical surveys in the region (Carter et al., 2007; Beem et al., 2017; Jordan et al., 2018) helps propagate englacial reflector ages throughout the region and add context to the new observations.

In this paper, we describe new basal topography and surface elevation, areas on the flanks of Titan Dome that may have previously experienced faster flow than at present, and that the basal ice age is likely younger than would be needed to capture the middle Pleistocene transition.

## 2 Data

### 2.1 New Data

The SPC survey was conducted by an aero-geophysical suite installed on a Polar Research Institute of China BT-67 airframe (Cui et al., 2018) that contains a coherent 60 MHz center frequency radar ice sounder (Peters et al., 2005), a laser altimeter, cesium magnetometer, three-axis stabilized gravimeter, and downward looking camera. The laser altimeter was a Riegl LD90-
3800-HiP and collected data at 4 Hz, with an expected accuracy of 15 cm. Two survey flights were conducted, in February of each 2016 and 2017 (fig. 1), over the area of Titan Dome that included coverage of a previously determined ice core target region (Van Liefferinge and Pattyn, 2013). A grid, roughly 150 km by 150 km with 25 km grid spacing, was surveyed. A survey line was flown within 500 meters of the South Pole Ice Core (Casey et al., 2014) to enable the propagation of the core's age model (Lilien et al., 2018) throughout the region.

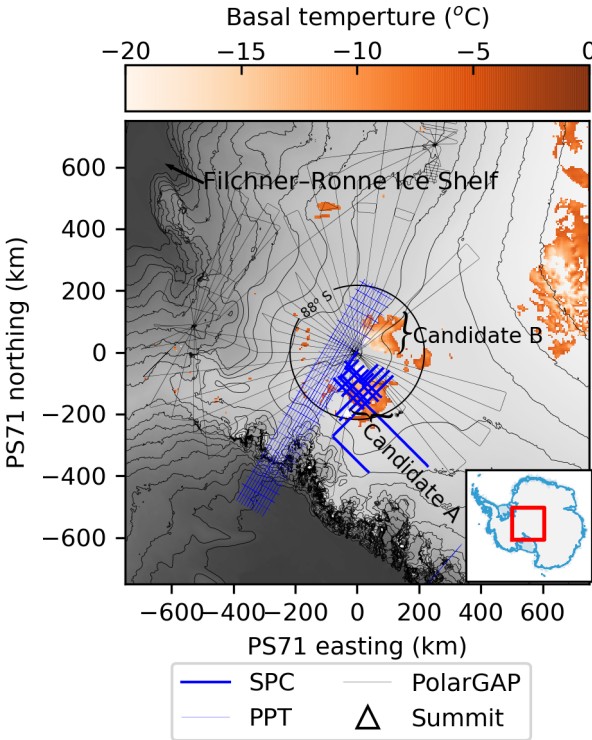

**Figure 1.** South Pole region. Flight lines from the South Pole Corridor (SPC), which is data presented here, and previously published observations of Pensacola-Pole Transect (PPT) from Carter et al. (2007) and PolarGAP (Jordan et al., 2018) surveys are plotted. The orange shading are regions of increased paleoclimate ice core potential plotted as basal temperature (Van Liefferinge and Pattyn, 2013). The two higher potential coring regions discussed in this paper are labeled Candidate A and B. The location of Titan Dome summit, as determined from the SPC survey, is the white triangle. The background shading and contours are from the Bamber et al. (2009) surface elevation DEM.

## 2.2 Existing Data

One older radar survey of the region is used in this analysis. The Pensacola-Pole Transect (PPT) was collected in 1998-1999. This data was collected on radar that was a direct ancestor of the system used for the South Pole Corridor survey. The PPT survey used a 60 MHz center frequency with 250 ns pulse width radar mounted on a Twin Otter airframe (Carter et al., 2007).





# 3 Methods

## 3.1 Radar Processing


The radar data was processed to a 1D focused state (Peters et al., 2007), without range migration. Focusing is applied to differentiate between nadir and off-nadir reflections and improve the resolution of the resulting radargram. The processing increases the discrimination of internal structure within and beneath the ice sheet.

The basal reflection coefficient has been corrected for energy loss due to the divergent beam pattern, also called geometric
spreading loss, and for assumed ice attenuation, which is primarily a function of ice temperature (MacGregor et al., 2007). Geometric spreading loss follows the standard theoretical relation using the infinite mirror approximation (e.g. Lindzey et al., 2020). Dielectric attenuation is reported as two way travel though a given ice thickness. A value of 10 dB km$^{-1}$ was used throughout the region. Although there is expectation that attenuation is variable due to spatial heterogeneity in ice temperature profile and/or ice chemistry, an attempt to constrain the variability is not made due to the numerous additional processes for
which a control would be needed (e.g. water distribution, geothermal flux heterogeneity, ice chemistry, basal roughness). The relative consistency of low magnitude basal reflection and the lack of inferred basal water, as will be described later (section 4.2), support the assumptions in determining the magnitude of the dielectric loss.

The attenuation value was determined by cross plotting basal reflectivity with ice thickness and regressing the distribution (fig. 2). Additional regressions that took a subset of the observations (thickness > 800, thickness > 1200) each resulted
in attenuation of 7 dB km$^{-1}$. To isolate the effects of dialectic loss within the ice column the highest or lowest values of reflectivity for a given ice thickness band can be used. In either case, the end member basal reflection coefficient is assumed to be consistent throughout the survey region and therefore isolates the effect of englacial attenuation. Specifically, the lowest and highest 5 values in each 50 m ice thickness bin were used. The number of values per bin has limited effect, changing the attenuation by only 2 dB km$^{-1}$ when using 1 to 10 values in each thickness bin. Regression of the lowest values within
each thickness bin (800 to 3000 m) results in an attenuation of 15 dB km$^{-1}$. The regression of the highest values results in 6 dB km$^{-1}$. As can be seen in fig. 2 the lowest value of reflectivity for the thinnest ice (500 to 1200 m) has a much steeper slope than the thicker ice. This could be due to the relative paucity of observations at these thicknesses. Ignoring the thinnest regions and regressing over 1200 to 3000 m of ice thickness and using the lowest values within each thickness bin results in attenuation of 11 dB km$^{-1}$. The same regression except with the highest values results in 9 dB km$^{-1}$. While some of these
analysis choices are arbitrary the attenuation is likely between 7 and 15 dB km$^{-1}$. Theoretical values of attenuation for -35°C ice, the approximate average temperature of a south polar ice column (Beem et al., 2017) is within the range 7 and 15 dB km$^{-1}$ range, depending on ice chemistry (MacGregor et al., 2007). These values are also consistent with the results of an ice sheet wide estimate of englacial attenuation (Matsuoka et al., 2012), which finds that the Titan Dome region has an attenuation consistent with the lower range of the estimates generated here. Consistent with theory and observations, 10 dB km$^{-1}$ is used.

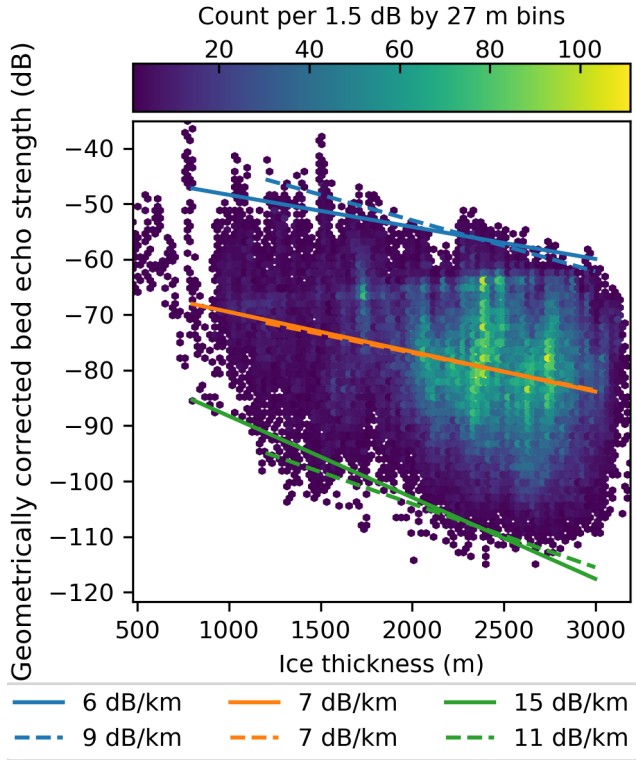

**Figure 2.** Attenuation determination. The color field represents the number of observations in each 27 m thickness bin and 1.5 db echo strength bin. The solid lines are regressions using observations with ice thickness greater than 800m, the dotted lines greater than 1200m. The orange lines use all observations in each thickness bin. The blue lines use the 5 highest echos strengths in each thickness bin and the green uses the 5 lowest.

## 3.2 Laser Altimetry

Laser altimetry was corrected for biases in the attitude of the sensor by minimization of the transect intersection differences (Young et al., 2015) with data from the 2015-2016 survey. As the laser and inertial navigation system was not removed from the aircraft between field seasons, recalibration was not required.

## 3.3 Surface, Bed, and Internal Isochron Tracing

The manual labeling of the surface and bed within the radar observations was consistent with the methodology described in Blankenship et al. (2001). The human labelers applied a first return criteria to label the horizons. This has the effect of identifying the minimum possible ice thickness and smoothing basal topography, especially in regions with steep and variable relief. Using the traced horizons along with aircraft position the surface elevation, bed elevation, and ice thickness are determined. Radar wave speed in ice is taken as $1.67 \times 10^8$ m s$^{-1}$.





Isochrons were manually traced using industry software. The South Pole Ice Core age model (Casey et al., 2014) was projected onto the radargram that flew most proximal to the core location (∼500 m), by correlating the ice depth of both the ice core and radar observations. Where isochrons are contiguous the age record can be propagated throughout the surveyed region. Nine age isochrons were propagated to their maximum possible extent from the South Pole Ice Core: 0 ka (taken as the surface), 4.7 ka , 10.7 ka, 16.8 ka, 29.1 ka, 37.6 ka, 51.4 ka, 72.5 ka, and 93.9 ka.

## 3.4    Basal Ice Age Model

The age of the basal ice can be modeled with the constraints from the dated isochrons. Two 1D models are compared to estimate the age of the basal ice. One model uses the simplest Nye assumptions which are a steady state ice thickness and a constant strain rate with depth (Cuffey and Paterson, 2010, eq. 15.8). For this model, vertical strain is only dependent on surface accumulation rate and basal ice is arbitrary defined as 30 meters above the bed. The model is solved for each vertical

record independently by finding the accumulation rate that minimizes the root mean squared error between the age model and the traced isochrons. The resulting accumulation field enables an estimate of the spatial patterns of average accumulation rate. Comparing the spatial distribution of accumulation from the model to observations of accumulation (Arthern et al., 2006; Wessem et al., 2014) serves as partial verification of the model.

The second age model makes the Dansgaard-Johnson set of assumptions concerning vertical strain rates (Cuffey and Pater-

son, 2010, eq. 15.13-15.15). In addition to setting an accumulation rate for the model, a characteristic height above the bed is set to mark the transition from constant vertical strain above to linearly varying to zero below. A range of transitional heights are tested, 0.2 to 0.5 of the ice thickness above the bed. This model is highly sensitive to accumulation rate, which sets the magnitude of vertical strain, but less sensitive to the chosen transitional height. Additionally, the definition of basal ice has a sensitive effect on the determined basal ice age, given the high degree of non-linearity this model produces in the deepest ice.

To improve on the arbitrarily defined 30 m above the bed, a minimum desired temporal resolution of ice, 10 kyr m$^{-1}$ (Fischer et al., 2013), is used to determine the basal ice age. The age when this temporal resolution threshold is exceeded is considered the basal ice age. The Dansgaard-Johnson model is solved independently for each vertical record by finding the accumulation that minimizes the model age and dated isochron root mean squared error misfit.

## 3.5    Submergence

Investigating the submergence rate, the length per unit time that a dated isochron takes to reach its current position, can be informative of the flow history in the region. Submergence rate is calculated in the same manner as Beem et al. (2017). We use the linear variability of strain rates, m=1 in eq 6 of Beem et al. (2017). Basically, the model assumes the form of the vertical strain rate profile and determines the magnitude of vertical strain necessary to submerge an isochron of a given age to its observed depth. Submergence rates are calculated for each dated isochron-bounded-interval within the ice column. There is a

correction step that removes the influence of the strain from each younger interval. Spatial patterns in submergence that exceed expected spatial gradients in accumulation are interpreted to represent spatial heterogenous basal melt or ice flow. The results


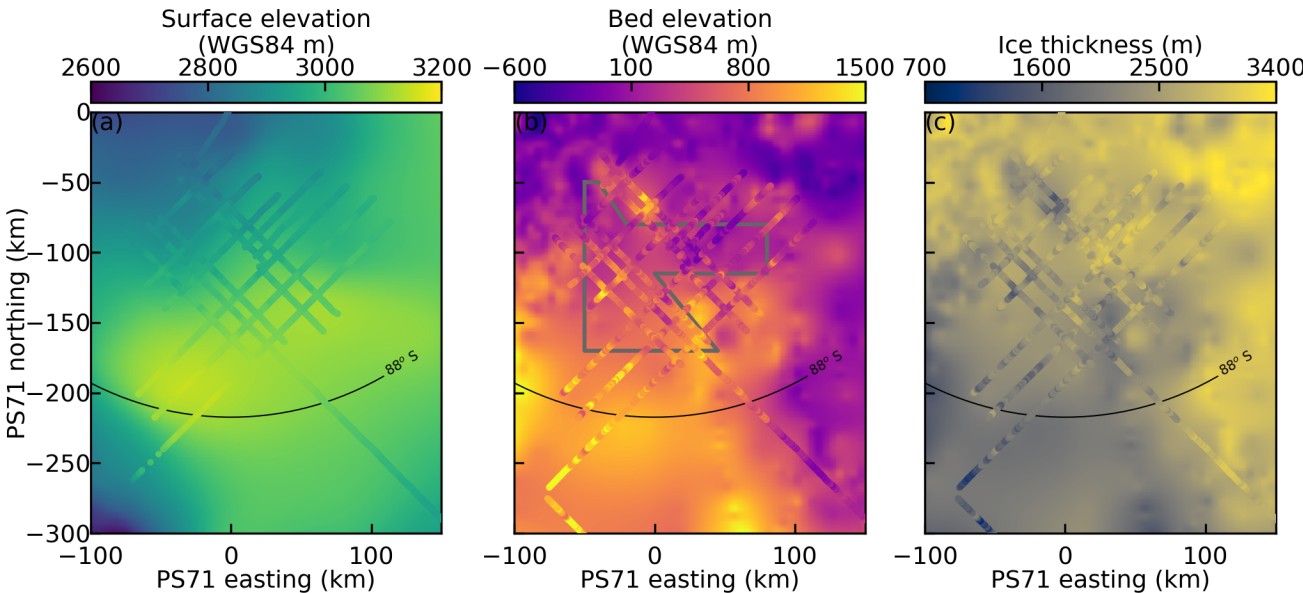

**Figure 3.** Basic observations. (a) Laser surface elevation with background of Bamber et al. (2009) DEM. (c) Radar derived bed elevation with a background of Bedmap2 (Fretwell et al., 2013). The polygon roughly trace subglacial troughs and are the same as in fig. 5. (e) Radar derived ice thickness with a background of Bedmap2 (Fretwell et al., 2013).

create a temporal history that is significant for interpreting the timing of any changes in processes that effect submergence rates (accumulation, basal melt, and/or horizontal strain).

## 4 Results

### 4.1 Bed and Surface Elevation and Ice Thickness

The bed and surface elevation observed by radar reflection constrain the location of the ice topographical high of the dome and reveal a mountainous subglacial terrain that was previously unknown. The bed topography includes multiple bedforms with over 1000 m of prominence (fig. 3 and 8).

There are two independent surface elevation DEMs of the Titan Dome region (Bamber et al., 2009; Helm et al., 2014). Other gridded DEM products (e.g. Bedmap2, REMA, BedMachine) use one of these two to fill in the data gaps south of 86° South (Fretwell et al., 2013; Howat et al., 2019; Morlighem et al., 2020), but can deviate from the source data due to the specific gridding and mosaicking implementation. Generally, there is good agreement between the available DEM products and the new laster altimetry observations of surface elevation (fig. 4). The Bamber et al. (2009) DEM is 20 +/- 62 m (average +/- 2 standard deviations) higher than radar observations and the Helm et al. (2014) DEM is 23 +/- 73 m higher. The Titan Dome





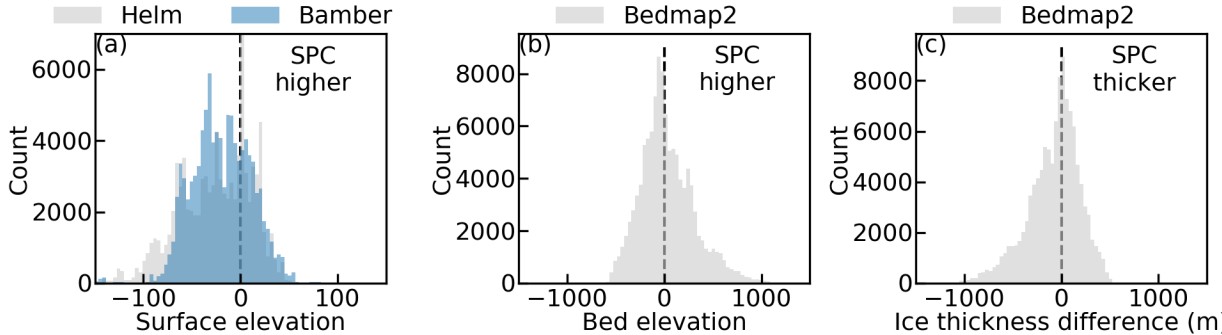

**Figure 4.** Difference between aerial observations and community DEMs. (a) Laser surface elevation difference from both Bamber et al. (2009) and Helm et al. (2014), (b) radar bed elevation observation difference from Fretwell et al. (2013), and (c) radar ice thickness observation from Fretwell et al. (2013).

summit location differs between the Bamber et al. (2009) and Helm et al. (2014) DEMs by at least 34 km. The aerial surface altimetry, collected here, is sparse and cannot explicitly constrain the location of the Titan Dome, but the dome location in the Bamber et al. (2009) was used in survey planning and is the location of highest elevation observed in this survey. The dome elevation is observed to be 3154 m and occurs at -88.1716° N, -99.5234° E, which is within 10 m of the Bamber et al. (2009) elevation and corresponds with their location of maximum surface elevation.

## 150    4.2    Basal Reflectivity

The bed beneath Titan Dome and the surrounding region show generally low reflectivity and heterogenous character. Isolated regions of higher values (> -30 db) are observed in the subglacial drainages that flows towards the Filchner-Ronne Ice Shelf and corresponds to basal topographic troughs (blue polygon in fig. 5). Higher values are also seen above the summit of the newly described subglacial mountain (blue circle in fig. 5) and an isolated location near the end of a survey line (green diamond 155   in fig. 5 near 100 km,-50 km PS71 coordinates).

    The distribution of basal reflectivity suggests that the basal ice beneath the dome is frozen to the bed with limited basal melt and water movement. This conclusion is consistent with previous basal temperature modeling efforts (e.g. Beem et al., 2017; Van Liefferinge and Pattyn, 2013; Price et al., 2002) that conclude the bed in the region is 10° C or more below the pressure melting temperature. The exception may be in the main drainage from the dome towards the Filchner-Ronne Ice Shelf which 160   shows higher reflectivity magnitudes, including some of the highest values observed in this survey (blue polygon in fig. 5). It is unlikely that pools of water were sampled, but the reflectivity suggests a higher likelihood of a smoother bed and/or small amounts of basal water in this region. The high reflectivity is seen near the summit of the subglacial high may be the result of the shallow ice conditions of this location and attenuation correction too large for the thin cold ice expected above this region.

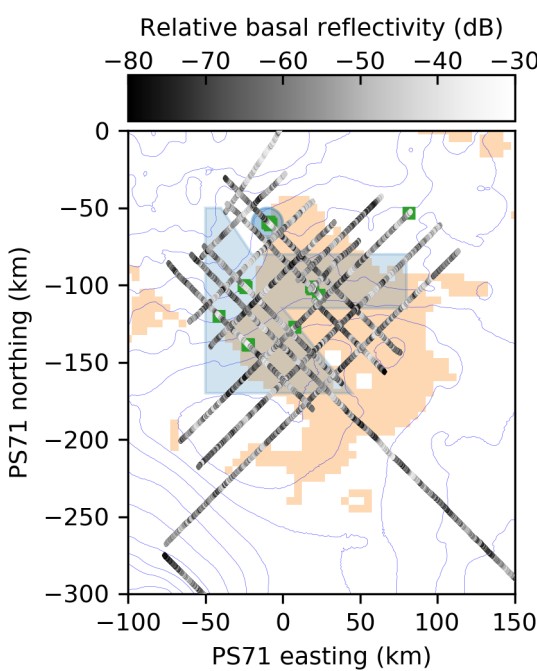

**Figure 5.** Observed relative basal reflectivity. Reflectivity is corrected for geometric spreading and englacial dielectric attenuation. Locations of reflectivity that exceed -30 dB are highlighted with green squares. The converging subglacial troughs are highlighted with blue polygon and the location the of subglacial mountain is the blue circle. Candidate A ice core target region is in orange (Van Liefferinge and Pattyn, 2013). The background 500 kPa contours are hydraulic potential using Bedmap2 (Fretwell et al., 2013) and zero effective pressure. The highest contour surrounding the dome summit is 29 kPa.

### 4.3 Basal Ice Age

Two age models were used to estimate the age of the basal ice, using the dated isochron as constraints. The Nye age model, predict basal ages as old as 350 ka with a mean accumulation (in ice equivalent) of 4.4 cm yr$^{-1}$. The spatial distribution of accumulation rates show lower magnitudes on the highest surface elevation of the dome and higher rates at lower elevations. This pattern and magnitudes are generally consistent with space-borne and re-analysis estimates of accumulation patterns of the region (Arthern et al., 2006; Wessem et al., 2014). This model result is not expected to be predictive of modern accumulation

rates, and there are regions that show higher magnitude than observations, however the general patterns are plausibly realistic and lend credence to the model performance despite its simplicity.

The Dansgaard-Johnson age model calculates older ages due to the model assumptions that include smaller magnitude vertical strain rates near the bed. Isolated regions exceeding 1 Ma of age are predicted to exist in the most favorable parameter

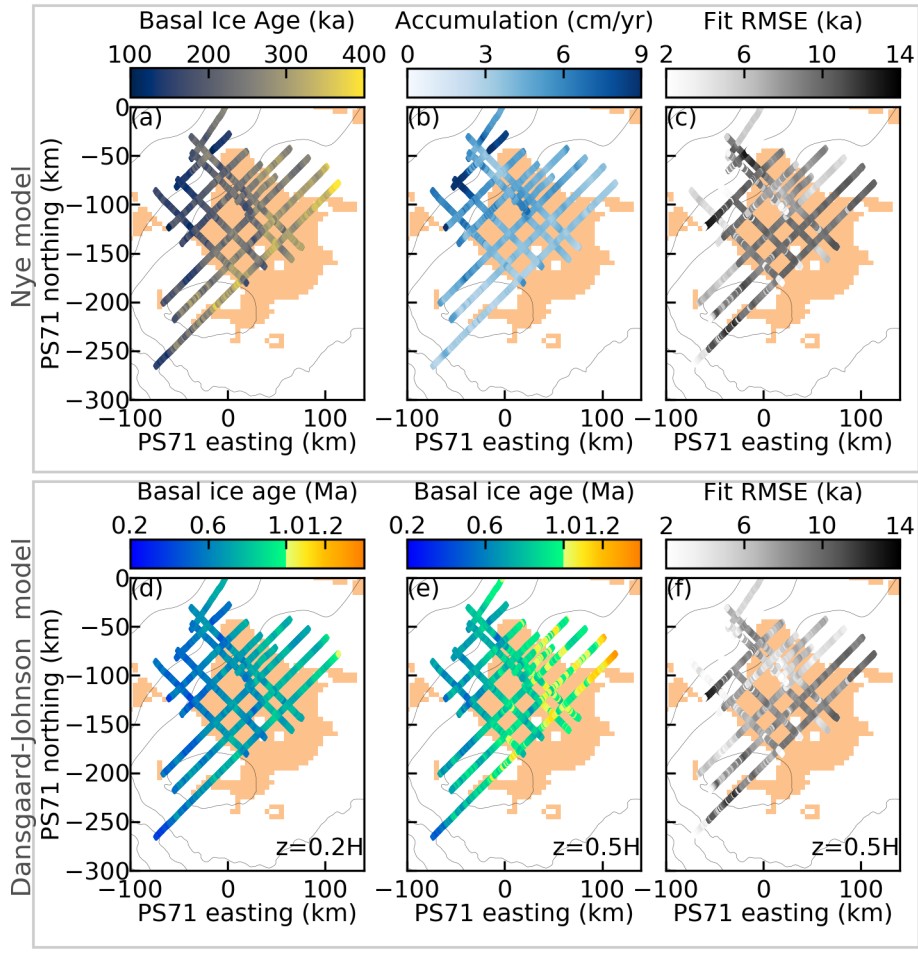

**Figure 6.** Age model results. The top row are the results of the Nye model assumptions and the bottom row the results of the Dansgaard-Johnson set of assumptions. The RMSE fit is the difference between the model and the traced and dated internal isochrons. Each panel is plotted over the higher probability candidate A ice core target region in orange (Van Liefferinge and Pattyn, 2013) and 100 m surface elevation contours (Helm et al., 2014). The dome is surrounded by the 3100 m contour.

sets, however ages between 600 and 800 ka are more typical. The higher the transitional height in the Dansgaard-Johnson

model the older the maximum basal age due to a greater proportion of the ice thickness with smaller vertical strain rates. With a transitional depth of 0.2 ice thickness the maximum age was ∼0.9 Ma and when the transition depth is 0.5 ice thickness the maximum age increases to greater than 1.4 Ma. Spatial variability in accumulation patterns and magnitudes were consistent with the Nye model results, less accumulation on the dome (∼2 cm yr$^{-1}$) and higher amounts on the flanks (up to 10 cm yr$^{-1}$). In every model case the probably of suitably old ice to capture of the middle Pleistocene transition is low.

At an ice divide, the Dansgaard-Johnson model is best applied with a transitional height that equals ice thickness, the strain rate varies linearly from the surface to the bed (Cuffey and Paterson, 2010, p. 619). In this scenario the basal ages become



considerably older, multiple millions of years. Although, such a strain rate profile is only relevant for a small portion of the survey, it creates a hypothetical that if the dome position was highly stable, the local conditions would create a suitable site for the extraction of an ice core that captures the middle Pleistocene transition. However, dome stability over these timescales is generally not expected, particularly for Titan Dome given its proximity to major and dynamic ice drainages (Trans-Antarctic mountain outlet glaciers and Filchner-Ronne ice streams) and the evidence that suggest the region has experienced more rapid flow in the past (Section 4.4).

## 4.4 Dated Isochron Depth and Submergence

Nine dated isochrons were traced to their maximum extent. The younger isochrons were traceable throughout the entire survey region, but older isochrons suffered from discontinuities and were increasingly limited in the extent in which they could be traced. Isochrons may have gaps in visibility due to dip steepness, being obscured by radar clutter, or ceasing to generate a suitably strong reflection for other reasons. The limited intersections of survey lines impede tracing around areas without isochron visibility. The 72.5 ka isochron was traced throughout a majority of the survey, but it was not possible to trace the 93.9 ka isochron beyond a few 10s of km from the ice core location.

The fractional depth of the 72.5 ka isochron ranges from 43% to 78% of the ice depth with a mean of 60% (fig. 6 and 8). The southern side of the dome show the deepest fractional depth for any given age isochron, as do regions near the prominent bedrock features (fig. 8). The observed depth of the 72.5 ka isochron significantly reduces the likelihood of sufficient temporal resolution if ice greater than 1 Ma old were to exist within the survey.

The submergence calculations put bounds on the timing of past ice flow deceleration. For the interval starting with the present, 0 to 4.7 ka, the gradients (fig. 7b) of submergence a similar to both the magnitude and pattern of present day accumulation (Arthern et al., 2006; Wessem et al., 2014), suggesting that this interval has been dominated by accumulation driven vertical strain. Similar to the conclusions of Beem et al. (2017), the submergence rates from the interval 10.7 to 16.8 ka (fig. 7c) show strong, but transitional, submergence boundaries. The 10.7 to 16.8 interval is the transition between the 4.7 to 10.7 ka submergence (not pictured) which is very similar to the 0 to 4.7 ka interval, and the 16.8 to 29.1 (fig. 7d) which also shows high gradient boundaries in submergence. This suggests the regional higher magnitude vertical strain ceased during the 10.7 to 16.8 ka interval. The new SPC observations show a contiguous region of greater submergence, isolated to a single ice catchment, that is bounded by higher gradients of submergence that are difficult to ascribe solely to accumulation patterns.

## 5 Discussion

Titan Dome has been previously identified (Van Liefferinge and Pattyn, 2013) as a region that holds potential for an ice core that would capture the middle Pleistocene transition (1.25 Ma to 0.7 Ma), due to slow flow, appropriate ice thickness, and the likelihood of basal temperatures that are well below the pressure melting point (Van Liefferinge and Pattyn, 2013; Beem et al., 2017; Price et al., 2002). The analysis completed here shows that the basal ice age is likely too young to be relevant for examination of the middle Pleistocene transition. While Titan Dome has ice of appropriate thickness, the dated isochron of

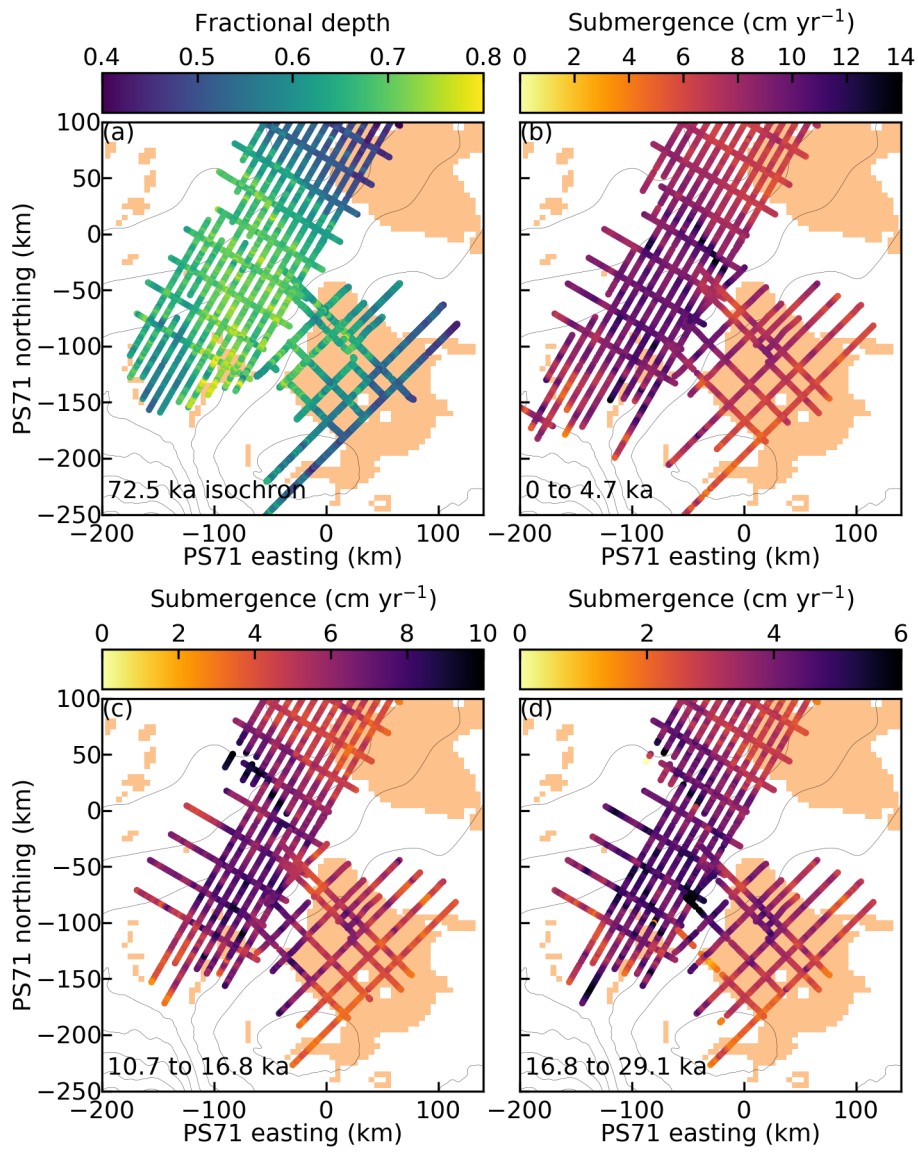

**Figure 7.** Dated isochron depth and submergence. The submergence panels, (b), (c), and (d), show the submergence calculated between the two dated isochrons. Each panel is plotted over orange shading, which are ice core target candidates A and B (Van Liefferinge and Pattyn, 2013), and 100 m surface elevation contours (Helm et al., 2014). The dome is surrounded by the 3100 m contour.

72.5 ka are at a significant fractional depth (50 to 70%), deceasing the likelihood of suitably old ice and severely limiting the
temporal resolution of old ice if it were to exist (fig. 7a). In comparison, Little Dome C of East Antarctica, has a 72 ka isochron modeled to be between 25% and 30% of the ice depth (Parrenin et al., 2017).

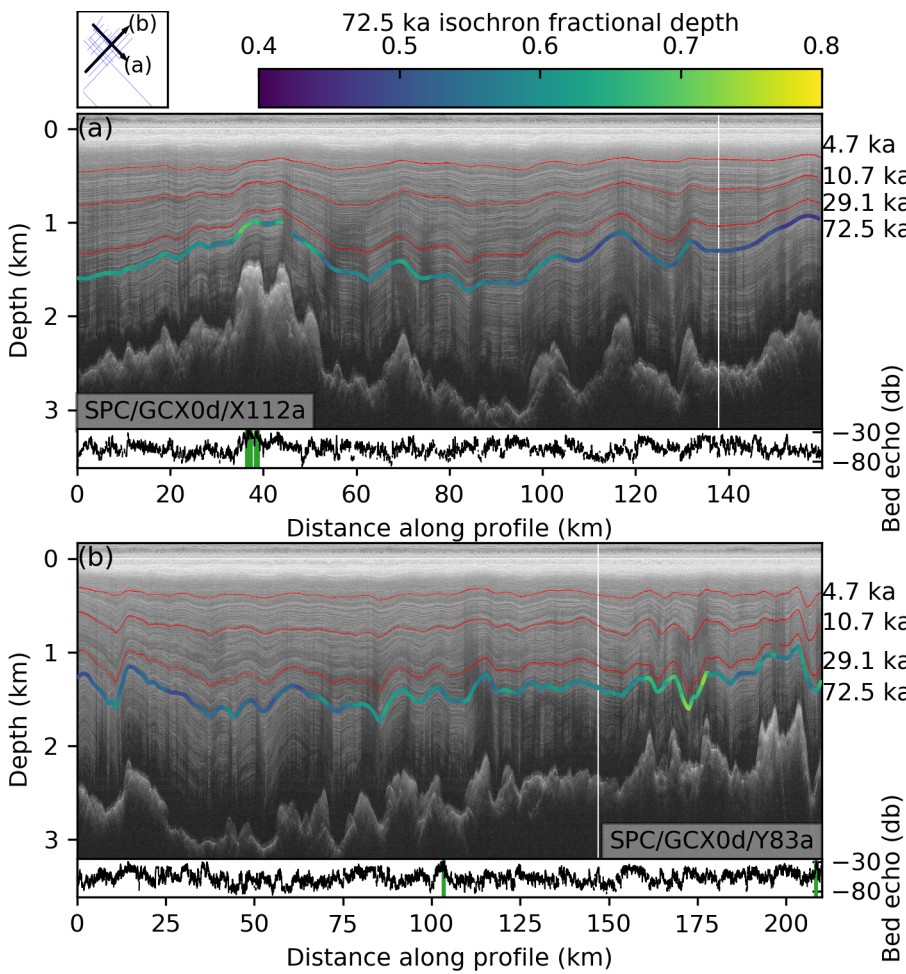

**Figure 8.** Example radargrams with traced isochrons. A context map is in the upper left corner. Beneath each panel is the bed echo strength, which when it exceeds -30 db is highlighted with vertical green lines, the same regions highlighted in fig. 5. The white vertical line on each radargram represents the crossover of the two transects.

The basal ice age models for Titan Dome fail to calculate ice of suitably old age. The Dansgaard-Johnson model, with assumptions that produce highly non-linear age approaching the bed, finds only isolated regions of 1.4 Ma in basal ice age. Although it is encouraging that the oldest modeled ages are on the dome and flanking ice divides, a typically suitable location for drilling an ice core, the specific locations with the oldest ages have been previously excluded from consideration due to exceeding the modern ice flow threshold of 2 m yr$^{-1}$.

The ice sheet modeling that identifies cold bedded drilling regions at Titan Dome (Van Liefferinge and Pattyn, 2013) used Bedmap2, which reports generally thicker ice than the SPC radar observations, Bedmap2 is 30 +/- 550 m (average +/- 2 standard deviations) thicker than the SPC radar observations. 50% of the radar observations within candidate A have thinner ice than Bedmap2 when linearly interpolated from the grid. Given the gridded nature of the modeling, 69% of the Bedmap2 pixels





within the promising area that were surveyed have thicker ice than the radar observations. The effect of thinner ice on the Van Liefferinge and Pattyn (2013) modeling is two fold and would have competing effects. Thinner ice would tend to increase the size of the promising region, because fewer locations would exceed their geothermal heat flux threshold for melting. Thinner ice would also decrease the extent of the promising region by increasing the balance velocities used to eliminate regions with
excessive horizontal ice advection. The Titan Dome region is more ice flow limited than temperature limited and the net effect of thinner ice would likely be a reduction in the extent of the promising region. For instance, the area between candidates A and B is excluded because it exceeds the balance velocity threshold ($>2$ m yr$^{-1}$). Additionally, some areas would be excluded as they are not thicker than the 2000 m threshold.

Titan Dome also shows evidence of increased ice flow in the past. This flow history could result in the loss of basal ice
through basal melting and complications in stratigraphic layering due to elevated strain rates. Recent publications have indicated past ice flow on the flanks of Titan Dome that is consistent with ice stream transitional flow (Beem et al., 2017; Lilien et al., 2018), that between slow ice deformation dominated flow in the interior ice sheet and that of basal slip dominated ice streaming. Isochron drawdown in the region is consistent with local melt from elevated geothermal flux, extensional strain, and/or frictional heating from past ice dynamics. The drawn down pattern, clearly evident in fig. 7(a), includes a linear bound-
ary that passes through PS71 50 km easting and 50 km northing and is completely within a single ice catchment. The linear boundary has previously been interpreted is a relic shear margin (Beem et al., 2017). The drawn down seen in the new SPC observations are consistent and contiguous with the pattern from the older Pensicola-Pole Transects observations.

Previous work (Beem et al., 2017) has suggested that the region decelerated between 10.7 and 16.8 ka. The submergence calculations from the SPC observations regions support the same conclusion that for a period ending between 10.7 and 16.8
ka the region experienced flow sufficient enough for vertical strain to be driven by horizontal ice flow gradients, such as from ice streaming or transitional flow. The ice streaming processes evident in drawdown of isochrons make the region more complicated for paleoclimate age model construction and the survivability of ice greater than 1 Ma.

The distribution of maximum basal echo strength locations is consistent with the regions of increased submergence, suggesting that this region may have experienced greater basal heating in the past. A hypothesis put forth in Beem et al. (2017)
suggests that remanent heat from past sliding may contribute to the present distribution of basal reflectivity and subglacial water. This hypothesis is generally supported by the new observations which show a contiguous region of increased submergence that is also has higher average basal reflectivity than the surrounding region. If this observation is representative of ice sheet dynamics, at least portions of the promising ice core target experienced faster flow in the past, which would decrease or eliminate its suitability for extracting an interpretable climate record, especially for one that extends to 1 Ma or beyond.
Candidate B (fig. 1) of modeled cold based ice, north along $\sim$45° meridian from South Pole, does not show enhanced flow history expressed through submergence rates. Using the same dated isochrons, but traced within the older Pensicola-Pole Transects data (Carter et al., 2007), the thickness of ice below the 93.9 ka isochron is generally less than 1000 m and in some instances less than 750 m. At least for the region with radar observations, it is unlikely that any ice greater than 1 Ma would have suitable temporal resolution.



The role or existence of elevated geothermal heat flux in the region is difficult to determine. Previous work has suggested a region proximal to Titan Dome has elevated geothermal flux (Jordan et al., 2018). However, the basal characterization of Titan Dome neither confirms or refutes the existence of elevated geothermal flux outside of the survey area. The new survey presented here is of a different subglacial catchment than that described in Jordan et al. (2018) and heterogeneity in geothermal flux is expected over length scales of 100s of kilometers. The basal characterization of Titan Dome does show a few areas with a higher likelihood for the existence of water. It is possible that localized geothermal flux could be causing the increased basal reflectivity, which may be in addition to or an alternative hypothesis to remnant heat from past basal sliding (Beem et al., 2017). These hypotheses could be tested through direct access of the bed to characterize the geology and measure geothermal flux.

## 6  Conclusions

Titan Dome is unlikely to be a suitable site for the extraction of ice for a climate proxy that captures the middle Pleistocene transition. The dated isochrons, the age models, and implications for faster flow in the past are each discouraging to the possibility of suitably old ice. Age models all indicate the basal ice age between 300 and 800 ka in the most promising locations. Older modeled ages do occur in some regions when using more favorable model parameters. In all instances, this is younger than the 1.25 Ma ice needed to study the complete middle Pleistocene transition. If 1 Ma or older ice were to exist, isochron ages, dated and propagated from the South Pole Ice Core (Casey et al., 2014), are too deep to have a suitable amount of ice for high temporal resolution. Further complication to any extracted ice core from Titan Dome is the evidence for faster flow in the past, which would distort chronology and source regions for any given layer within the core.

The new observations also described previously unknown basal topography, including a large subglacial mountain. The new observations can be used to further improve community data sets in this region, they are already within the BedMachine product (Morlighem et al., 2020). The location of the Titan Dome summit is observed to be consistent with the Bamber et al. (2009) ice surface DEM at -88.1716° N, -99.5234° E.

*Data availability.* The L2 data is made available at USAP data portal...

*Author contributions.* This project was made possible by funding acquired by DDB, DAY, JG and SB and project administration by DAY, DDB, JG, SB. Conceptualization of the project was completed by LHB, DAY, and JSG with investigation by JSG. Formal analysis, methodology, and visualization was completed by LHB. LHB wrote the original draft and all authors contributed to review and editing. Data curation was completed by DAY and LHB.

*Competing interests.* The authors declare they have no conflict of interest.



*Acknowledgements.* The authors would like to thank Polar Research Institute of China for their support and making their aerial geophysical platform available for this research. The authors also thank Ken Borek Ltd. pilots and engineers for the involvement and support in data col-

lection. The authors thank Mercy Grace Browder and Roccio Castillo for their efforts in radar interpretation and the support of the Jackson School for Geoscience GEOFORCE program. The work here was supported by NSF Grant #1443690 (SPICECAP), G. Unger Vettlesen Foundation, the National Natural Science Foundation of China (41876227) and the National Key R&D Program of China (2018YFB1307504). This is UTIG contribution #XXXX.



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
