# Peer review of "Aerogeophysical characterization of Titan Dome, East Antarctica, and potential as an ice core target"

_The Cryosphere, 2020_

## Referee Comment (RC1) · Massimo Frezzotti (Referee) · 30 Aug 2020

This manuscript provides new aerial geophysical observations allowing to constraints a basal ice age modelling on the flanks of Titan Dome. This Dome was previously identified as a contender for possible deep ice core site that could capture the middle Pleistocene transition (900-1200 kyrs).

Titan Dome (88.50° S, 165.00° E) is located about 200 km from South Pole and It was delineated by the SPRI-NFS-TUD airborne radio-echo sounding program between 1967 and 1979. The Dome and its southern flanks are beyond the geographic limit of many satellite-based observations (e.g. ICESat 86°S; SSM/I 87°S; CryoSat-2, 88°S) and only few old air-borne radar surveys provide data of surface elevation, ice thickness

and bed topography.

The requirements for a site to collect stratigraphically intact oldest ice core are: low snow accumulation, low geothermal heat flow, proximity to an ice dome/divide, limited basal roughness, ice thicknesses of about 2500-2700 m (Fisher et al. 2017).

Acquisition of an accurate ice age modelling on the base of detail geophysical observation is prerequisite for any paleoclimatic ice core site selection. Authors used new and previous radio-echosounding and laser altimeter data to provide new surface elevation, ice thickness and bedrock topography.

The important effort to acquire new geophysical information in remote area and their analysis must be supported and the main results are of interest, but someone already published in Beem et al., 2017. However, this manuscript suffers of some flaws, in particular:

both the age model use snow accumulation value, but the authors do not provide sufficient information on spatial variability in the analysed area and their source data (e.g. Fig.2 snow accumulation map on the base of Arthern or Wessen with background the snow accumulation derived by 4.7 kyr isochrone), the source of snow accumulation value are not everywhere clarified and made the manuscript difficult to follow; the snow accumulation of 4.4 cm/yr i.e. are not explained, and not clarify if represent the present snow accumulation or the mean value snow accumulation rate histories taking in account the reduction during glacial period and how is calculated;

the source/process of isochronal layer ages are not explained and does not take in account the recent result of SPICEcore (e.g. Winski et al., 2019) everywhere;

the analysis of Candidate B site is reduced at 5 lines on discussion, remove the B site or analyzed in more detail;

the comparison with the previous DEM does not take in account the source of the data (SPRI-NFS-TUD airborne radio-echo sounding) southern of 88°S. The uncertain and

accuracy in elevation and position of the data southern of 88°S are very different from altimetry satellite data;

the RES tracks of BedMap2 must be shown in figure 2 and analyzed a used for provide new maps;

the suggest change in ice velocity from Beem et al. 2017 have impact on dome/ice divide position, the authors should analyze this point in relation with potential ice core site, column stratigraphy integrity and upstream correction, more than report the previous results;

the proposed geographical coordinates of the Titan Dome position are in an unusual format (88.1716° N, -99.5234° E) and longitude is wrong (line 148 and 281). The longitude value must be correct everywhere.

In detail: Line 22 and 234: the threshold of ice thickness in Fisher et al. 2013 is much higher than 2000 m for snow accumulation of 4.4 cm/yr i.e., and ice velocity lower;

line 23: add the minimum desired temporal resolution of ice of 10 kyr m−1, it is very important the resolution at MPT to resolve 41 kyr cycle;

Line 36: Titan Dome position is along around 160°E meridian;

Fig. 1 the summit position proposed is not visible, the PPT line of legend is too light;

Line 85: the theoretical value of attenuation of -35°C is not reported in Beem et al. 2017, provide more information;

Fig. 3: add elevation contour line with value and proposed dome summit, add new panel with snow accumulation map with accumulation derived by 4.7 kyr isochrone, a new panel with ice thickness difference;

Paragraph 4.1 and fig. 4 see general comments;

Fig.5 Add proposed summit position and the value of hydraulic potential contour;

Paragraph 4.3 see general comments on snow accumulation;

Fig. 6: add summit position and elevation contour value; For Dansgaard-Johnson model is not correct the label of panel as Basal ice age because the modelled age is at z=0.2H and 0.5H;

Line 178-179 Please explain how the spatial variability of snow is consistent with Nye Model, it is a circular reasoning;

Paragraph 4.4, fig.7 does not show clearly the difference in submergence velocity, improve the color scale;

Fig. 7 The first panel fractional depth is not described;

Line 225-226 see general comments on previous radarsounding data;

Line 231-232 why the area between the two-candidate area were exclude? Explain the source of ice velocity >2m/yr, surface elevation morphology and ice thickness are not different in the area;

Line 232 please explain the 2000 m of ice thickness threshold;

Line 234 Titan Dome could be migrated in the past, the evidence of change in ice velocity has been proposed for the flank about 100 km far, not at dome site;

Line 236 the pattern is not clear evident in fig. 7a;

Line 255 candidate B site is forward to Dome A from South Pole, along the 80° meridian East;

Line 264 The subglacial catchment basin analysed by Jordan et al. 2018, include completely the candidate A and B area, rephrase.

---

## Referee Comment (RC2) · Neil Ross (Referee) · 9 Oct 2020

Review of: Characterization of Titan Dome, East Antarctica, and potential as an ice core target (MS No.: tc-2020-210)

This manuscript provides a detailed and in-depth analysis of Titan Dome, East Antarctica. The manuscript characterises the study area using radio-echo sounding data and modelling to characterise the englacial layering and basal properties of Titan Dome, using these to assess its viability as an 'old-ice' target. The manuscript makes a substantive and important contribution to the discipline and study region.

The manuscript is reasonably well-written, the datasets and methods are comprehensively reported, and the data and modelling are presented reasonably effectively

(though please see my comments below and in the annotated manuscript).

General comments: There are a few general issues that I believe are worth highlighting about the manuscript in its current form. Were these to be addressed, the manuscript would be much improved:

1. A number of the figures require improvement to present the data effectively and to give the reader a fuller appreciation of 'Titan Dome'. 2. An improved description of the datasets (e.g. ice thickness, bed topography) are required in section 4.1. 3. An improved characterisation of the locational context of the study is required. There are numerous references in the text to "ice catchments" but these are never named (e.g. Academy Glacier, Patuxent Ice Stream etc.). 4. Engagement with relevant recent literature should be improved. Papers by Winter et al. 2018 https://doi.org/10.1029/2018GL077504, Paxman et al. 2019 https://doi.org/10.1029/2018GC008126 and Studinger et al 2020 https://doi.org/10.5194/tc-14-3287-2020 may help to provide more context (e.g. for naming ice catchments etc.), and for placing Titan Dome in a wider geographical, glaciological and geophysical context - the authors may wish to consider using some of these papers for a more developed study area section, and may wish to integrate the Studinger et al. 2020 paper into their discussion on surfaceaccumulation?

5. There are quite a few grammatical errors throughout the text that will need rectified.

6. The authors should consider adding to the introduction a short section describing the use and modelling of englacial layering for developing age stratigraphies for the ice sheet. There are a number of recent papers (e.g. https://essd.copernicus.org/articles/11/1069/2019/ or https://doi.org/10.1029/2019GL086663) that would be relevant, in addition to the papers of Marie Cavitte that several of the authors of this manuscript were also involved with.

Specific comments on figures Figure 1 – A zoom in of Titan Dome is required. -

[Figure]

define PS on the axis labels - it is unclear why Filchner Ronne Ice Shelf is annotated, but Ross Ice Shelf is not. Labelling of major outlet glaciers and/or glaciological catchments would be useful. - typo: "temperture" - authors should consider improving the colour scale for the basal temperature - the plotting of the polargap survey lines are (a) difficult to make out; and (b) incomplete – see figure 1 of https://agupubs.onlinelibrary.wiley.com/doi/full/10.1029/2018GC008126 - there is no need to have 'candidate' written on the figure. A and B will be sufficient.

Figure 3 - the new data presented in this figure are generally rather lost in the background. What about plotting the new along track data without the underlying DEMs, and then having a set of line data vs. DEM difference figures (e.g. from Bedmap2, Bamber DEM etc.) - Contours would be useful where DEM grids are shown. - later in the manuscript there is numerous references to a new subglacial mountain. Please annotate this in 3b.

Figure 5 - The way that the reflectivity data have been plotted in this figure makes it indecipherable, at least to me. Authors should look to improve the display of data in this figure. - Subglacial Troughs: these need described in section 4.1 of the paper, and they also need to be represented better in this figure. The blue polygon looks nothing like two troughs to me.

Figure 6 - this figure is afflicted with the same problems of figure 5. It is very difficult to make out the detail given the colour scheme and the way in which the data have been plotted.

For further specific comments on the manuscript, please see annotated PDF attachment.

Dr Neil Ross Newcastle University 9th October 2020

Please also note the supplement to this comment:
https://tc.copernicus.org/preprints/tc-2020-210/tc-2020-210-RC2-supplement.pdf

[Figure]

[Figure]

**Supplement:**

**Characterization** of Titan Dome, East Antarctica, and potential as an ice core target**

Lucas H. Beem1, Duncan A. Young2, Jamin S. Greenbaum2, Donald D. Blankenship2, Jingxue Guo3, and Sun Bo3

[revised manuscript text omitted]

---

## Author Comment (AC1) · 16 Jan 2021

**Review Response for: Characterization of Titan Dome, East Antarctica, and potential as an ice core target**

**1 General Comments**

**1.1 RC1: somewhat already published in Beem et al., 2017**

We disagree with this characterization. The work in this manuscript is not a republishing of past work. This manuscript supplies new observations that support numerous hypothesis. The most important of which is that ice age at this location is too young to study the middle Pliestocene transition. Also relevant and of interest, the observations support the hypothesis proposed in (Beem et al., 2017) and those ramifications are addressed in the discussion section. We think it well within the typical practice to build upon past work, even our own, and discuss the new observations within the context of past work.

**1.2 There are numerous references in the text to "ice catchments" but these are never named (e.g. Academy Glacier, Patuxent Ice Stream etc.).**

The "ice catchment" nomenclature has been simplified which obviates the need for geographical place names.

**1.3 Engagement with relevant recent literature should be improved. Papers by Winter et al. 2018 https://doi.org/10.1029/2018GL077504, Paxman et al. 2019 https://doi.org/10.1029/2018GC008126 and Studinger et al 2020 https://doi.org/10.5194/tc-14-3287-2020 may help to provide more context (e.g. for naming ice catchments etc.), and for placing Titan Dome in a wider geographical, glaciological and geophysical context - the authors may wish to consider using some of these papers for a more developed study area section, and may wish to integrate the Studinger et al. 2020 paper into their discussion on surface accumulation?**

Thank you for these suggestions. Citations have been added.

**2 Title/Abstract**

**2.1 RC2: change title to "Aerogeophysical Characterization..."**

This is a fine suggestion and makes the title more informative. We have made the change.

**2.2 RC2: a rather unusual opening sentence for an abstract? I'm not sure that the geographical information is that necessary in the abstract. It is detail that can be provided later.**

Geographical context seems like a solid place to start but "along a meridian" might be superfluous at this moment and has been removed.

**2.3 RC2: suggestions for additions or wording change**

These were helpful and improvements were made to the abstract. Thank you.

**3 Introduction**

**3.1 RC1: Line 22 and 234: the threshold of ice thickness in Fisher et al. 2013 is much higher than 2000 m for snow accumulation of 4.4 cm/yr i.e., and ice velocity lower;**

In both instances, the 2000 m threshold was discussed in regard to previous work (i.e. Van Liefferinge and Pattyn, 2013). We have changed the language of the sentences to make it clearer we were discussing the work of others that used this specific threshold.

**3.2 RC1:add the minimum desired temporal resolution of ice of 10 kyr/m, it is very important the resolution at MPT to resolve 41 kyr cycle**

We agree the temporal resolution is significant for any 'old ice' core. This detail is discussed later in the paper. It was not included here, as this paragraph described the motivating work of Van Liefferinge and Pattyn (2013). Since these other authors did not considered this threshold explicitly in determination their potential coring regions, it is not discussed here.

**3.3 RC1: Titan Dome position is along around 160°E meridian**

It is accurate for us to use -170 °E. We have made that change.

**3.4 RC2: The authors should consider adding to the introduction a short section describing the use and modelling of englacial layering for developing age stratigraphies for the ice sheet. There are a number of recent papers (e.g. https://essd.copernicus.org/articles/11/1069/2019/ or https://doi.org/10.1029/2019GL086663) that would be relevant, in addition to the papers of Marie Cavitte that several of the authors of this manuscript were also involved with.**

Thank you for these suggestions. Citations have been added.

**4   Methodology**

**4.1 RC1: both the age model use snow accumulation value, but the authors do not provide sufficient information on spatial variability in the analyzed area and their source data (e.g. Fig.2 snow accumulation map on the base of Arthern or Wessen with background the snow accumulation derived by 4.7 kyr isochrone), the source of snow accumulation value are not everywhere clarified and made the manuscript difficult to follow; the snow accumulation of 4.4 cm/yr i.e. are not explained, and not clarify if represent the present snow accumulation or the mean value snow accumulation rate histories taking in account the reduction during glacial period and how is calculated.**

We point the reviewer to lines 108-111 of the "discussion" article where we state: "The [Nye] model is solved for each vertical record independently by finding the accumulation rate that minimizes the root mean squared error between the age model and the traced isochrons." The accumulation rate is the result of the model.

In the results section, where 4.4 cm/yr is reported we changed the language to make it clearer that this is the output of the Nye model and necessarily assumes a steady state accumulation rate. Within that same paragraph we discuss how these magnitudes should be viewed, as rough approximations of present day accumulation given the simplicity of the Nye model, but that spatial patterns and magnitudes are qualitatively similar to available surface accumulation datasets.

**4.2 Line 85: the theoretical value of attenuation of -35°C is not reported in Beem et al. 2017, provide more information;**

Attenuation is reported in MacGregor et al. (2007). Beem et al. (2017) supports the contention of the average ice column temperature. We have rewritten the sentence to more clearly delineate what information can be found in the associated citation.

**5 Results**

**5.1 RC1:the source/process of isochronal layer ages are not explained and does not take in account the recent result of SPICEcore (e.g. Winski et al., 2019) everywhere;**

We have taken into account Winski et al. (2019). We used that data set in our analysis. It was inaccurately cited. We have fixed that oversight.

We find it beyond the scope of this paper to discuss ice core dating methodology.

**5.2 RC1: the comparison with the previous DEM does not take in account the source of the data (SPRI-NFS-TUD airborne radio-echo sounding) southern of 88∘S. The uncertain and accuracy in elevation and position of the data southern of 88∘S are very different from altimetry satellite data;**

Neither of the compared surface elevation DEMs (Bamber et al., 2009; Helm et al., 2014) used airborne echo sounding to determine surface elevation. Considering the uncertainties differences between airborne and space-borne observations is not necessary.

**5.3 RC1: the proposed geographical coordinates of the Titan Dome position are in an unusual format (88.1716∘ N, -99.5234∘ E) and longitude is wrong (line 148 and 281). The longitude value must be correct everywhere.**

There was an error in the coordinate. The correct coordinate reporting in the convention of using North and East coordinates, the values should have been -88.1716 °N and -170.4765 °E. These values have been corrected.

**5.4 RC1: Please explain how the spatial variability of snow is consistent with Nye Model, it is a circular reasoning;**

The reviewer appears to believe that the accumulation rate is supplied a priori as an input to the model. Instead, accumulation is the value that the model solves for upon fitting to dated internal horizons. Comparing the output of the model to independently collected observations is a means to validate the model results. We have made changes to the text, in both the methodology and results, to enable readers to more easily understand our work.

**5.5 RC2: An improved description of the datasets (e.g. ice thickness, bed topography) are required in section 4.1.**

An additional paragraph expanding the description of the bed/ice thickness has been added.

**6  Discussion**

**6.1  RC1: the analysis of Candidate B site is reduced at 5 lines on discussion, remove the B site or analyzed in more detail**

Although Candidate B is tangential to the new observations, we believe the little information we have concerning Candidate B is still of relevance to the scientific community's search for future drilling sites.

**6.2  RC1: the suggest change in ice velocity from Beem et al. 2017 have impact on dome/ice divide position, the authors should analyze this point in relation with potential ice core site, column stratigraphy integrity and upstream correction, more than report the previous results;**

To analyze column stratigraphy integrity and up stream correction would, of course, be of significant interest. However, to do this properly would require isochron tracking 3D ice sheet model experiments which far exceeds the scope of this paper. We hope that this publication would serve as possible motivation for this work to occur in the future.

What can be said based on our results in that the Titan Dome region shows contiguous and regional internal structure draw down which is consistent with the region experiencing faster flowing ice. We suggest the hypothesis that such faster flow could have impacted the ice divide geometry, moving it further interior. Such an ice sheet evolution history would be important to account for in any future ice core drilled in the location, and may disqualify such a location from consideration of a 1.5 million year old core.

**6.3  RC1: Line 231-232 why the area between the two-candidate area were exclude? Explain the source of ice velocity >2m/yr, surface elevation morphology and ice thickness are not different in the area;**

The text here as been updated to make it clear that the region is removed by the authors Van Liefferinge and Pattyn (2013) and their work. This citation does not specifically report their ice velocity results. But the work of Bingham et al. (2007) finds enough difference in this area to calculate higher balance velocities for exactly the region between what we are calling candidate A and B. The additional citation is now included.

**6.4  RC1:Line 232 please explain the 2000 m of ice thickness threshold;**

New text has been added to make it clear that this threshold was used to determine the regions of cold bedded glacier defined by Van Liefferinge and Pattyn (2013). The Van Liefferinge and Pattyn (2013) work motivated the aerogeophysical survey that is reported in this paper.

**6.5 RC1: Line 234 Titan Dome could be migrated in the past, the evidence of change in ice velocity has been proposed for the flank about 100 km far, not at dome site;**

"Flanks" has been added to the text.

**6.6 RC1: Line 236 the pattern is not clear evident in fig. 7a;**

I understand that the spatial transition being discussed is a bit diffuse. But the linear feature in isochron draw down in figure 7a and 7d are difficult to explain by slow continental ice flow alone. We have added arrows to figure 7a and 7d to draw the eye to the region.

**6.7 Line 255 candidate B site is forward to Dome A from South Pole, along the 80° meridian East;**

Given its size and location, candidate B intersects both 45 °E and 80 °E. We have left the value as is.

**6.8 Line 264 The subglacial catchment basin analysed by Jordan et al. 2018, include completely the candidate A and B area, rephrase.**

We have made it clearer that we are referring to only the region of higher geothermal flux identified by Jordan et al. While the full data set (PolarGAP) used by Jordan et al. (2018) does encompass numerous catchments including the ones that flow from Titan Dome. The region that they identified as warmer is in a different ice catchment than that which hosts the majority of our SPC survey.

**7 Figure 1**

**7.1 RC1: Fig. 1 the summit position proposed is not visible, the PPT line of legend is too light;**

Figure 1 has been remade.

**7.2 RC2: A zoom in of Titan Dome is required. define PS on the axis labels - it is unclear why Filchner Ronne Ice Shelf is annotated, but Ross Ice Shelf is not. Labelling of major outlet glaciers and/or glaciological catchments would be useful. - typo: "temperture" - authors should con- sider improving the colour scale for the basal temperature - the plotting of the polargap survey lines are (a) difficult to make out; and (b) incomplete – see figure 1 of https://agupubs.onlinelibrary.wiley.com/doi/full/10.1029/2018GC008126 - there is no need to have 'candidate' written on the figure. A and B will be sufficient.**

Figure 1 has been remade.

**8 Figure 3**

**8.1 RC1: Fig. 3: add elevation contour line with value and proposed dome summit, add new panel with snow accumulation map with accumulation derived by 4.7 kyr isochrone, a new panel with ice thickness difference;**

Age model results including accumulation can already be found in figure 6. Ice thickness differences can be found more in the remade figure 3 and in figure 4.

**8.2 RC2: the new data presented in this figure are generally rather lost in the background. What about plotting the new along track data without the underlying DEMs, and then having a set of line data vs. DEM difference figures (e.g. from Bedmap2, Bamber DEM etc.) - Contours would be useful where DEM grids are shown. - later in the manuscript there is numerous references to a new subglacial mountain. Please annotate this in 3b.**

Figure 3 has been reorganized to included difference maps, which both reviewers suggested. The subglacial mountain has been removed as a point of reference, so it is not necessary to include in a figure.

**9 Figure 5**

**9.1 RC1: Fig.5 Add proposed summit position and the value of hydraulic potential contour;**

The summit position has been added to figure 3 for clearer determination by the reader. Although I am usually a proponent of direct labeling on figures. In this instance, the contour labels clutters the image. As the values of hydropotential are less significant than the gradients, the contours without labels communicate the important information. The contour values are included within the figure caption for any reader who is interested in their magnitude.

**9.2 RC2:The way that the reflectivity data have been plotted in this figure makes it indecipherable, at least to me. Authors should look to improve the display of data in this figure. - Subglacial Troughs: these need described in section 4.1 of the paper, and they also need to be represented better in this figure. The blue polygon looks nothing like two troughs to me.**

The blue polygon (and the discussion of the troughs) has been removed which reduces visual clutter. The colored lines representing basal reflectivity magnitude have been made wider for easier visual discrimination. The colormap has also been adjusted.

**10 figure 6**

**10.1 RC1: add summit position and elevation contour value; For Dansgaard-Johnson model is not correct the label of panel as Basal ice age because the modelled age is at z=0.2H and 0.5H;**

The summit position in now in figures 1,3, and 5. The elevation contours of the dome are not the important details of this figure. The elevation contours are used for general geographic context. As a way to reduce visual clutter (which RC2 has asked to be addressed), the contour values have been omitted. An interested reader can determine contour values from information in the figure caption.

Our use of z was unnecessarily confusing. We have changed the variable to h and included equations to define the variables used. Text, figures, and figure captions have been updated.

**10.2 RC2: this figure is afflicted with the same problems of figure 5. It is very difficult to make out the detail given the colour scheme and the way in which the data have been plotted.**

The gray colorbar has been switched for a brighter map and a gray background was selected to increase contrast. Perhaps the black to white colorbars of figure 5 and panels 6c and 6f were causing the main difficulty in detail discrimination. We hope these changes makes the figures easier for readers to digest.

**11 figure 7**

**11.1 RC1: fig.7 does not show clearly the difference in submergence velocity, improve the color scale;**

A gray background and updated colormaps have been used to try to increase the visibility of the figure's detail.

**11.2 RC1: The first panel fractional depth is not described;**

The reviewer is correct. The figure caption has been updated.

**References**

Bamber, J., Gomez-Dans, J., and Griggs, J.: A new 1 km digital elevation model of the Antarctic derived from combined satellite radar and laser data–Part 1: Data and methods, The Cryosphere, 3, 101–111, https://doi.org/10.5194/tc-3-101-2009, 2009.

Beem, L. H., Cavitte, M. G. P., Blankenship, D. D., Carter, S. P., Young, D. A., Moldoon, G., Jackson, C. S., and Siegert, M. J.: Ice-flow reorganization within the East Antarctic Ice Sheet deep interior, in: Exploration of Subsurface Antarctica: Uncovering Past Changes and Modern Processes, edited by Siegert, M. J. and Jamieson, S. R. abd White, D. A., vol. 461, Geological Society of London, https://doi.org/10.1144/SP461.14, 2017.

Bingham, R. G., Siegert, M. J., Young, D. A., and Blankenship, D. D.: Organized flow from the South Pole to the Filchner-Ronne ice shelf: An assessment of balance velocities in interior East Antarctica using radio echo sounding data, Journal of Geophysical Research: Earth Surface, 112, F03S26, https://doi.org/10.1029/2006JF000556, 2007.

Helm, V., Humbert, A., and Miller, H.: Elevation and elevation change of Greenland and Antarctica derived from CryoSat-2, The Cryosphere, 8, 1539–1559, https://doi.org/10.5194/tc-8-1539-2014, 2014.

Jordan, T., Martin, C., Ferraccioli, F., Matsuoka, K., Corr, H., Forsberg, R., Olesen, A., and Siegert, M.: Anomalously high geothermal flux near the South Pole, Scientific reports, 8, 16 785, https://doi.org/10.1038/s41598-018-35182-0, 2018.

MacGregor, J. A., Winebrenner, D. P., Conway, H. B., Matsuoka, K., Mayewski, P. A., and Clow, G. D.: Modeling englacial radar attenuation at Siple Dome, West Antarctica, using ice chemistry and temperature data, Journal of Geophysical Research: Earth Surface (2003–2012), 112, F03 008, https://doi.org/10.1029/2006JF000717, 2007.

Van Liefferinge, B. and Pattyn, F.: Using ice-flow models to evaluate potential sites of million year-old ice in Antarctica, Climate of the Past, 9, 2335–2345, https://doi.org/10.5194/cp-9-2335-2013, 2013.

Winski, D. A., Fudge, T. J., Ferris, D. G., Osterberg, E. C., Fegyveresi, J. M., Cole-Dai, J., Thundercloud, Z., Cox, T. S., Kreutz, K. J., Ortman, N., et al.: The SP19 chronology for the South Pole Ice Core–Part 1: volcanic matching and annual layer counting, Climate of the Past, 15, 2019.